# THE BENEFIT OF DISTRACTION: DENOISING REMOTE VITALS MEASUREMENTS USING INVERSE ATTENTION

## ABSTRACT

Attention is a powerful concept in computer vision. End-to-end networks that learn to focus selectively on regions of an image or video often perform strongly. However, other image regions, while not necessarily containing the signal of interest, may contain useful context. We present an approach that exploits the idea that statistics of noise may be shared between the regions that contain the signal of interest and those that do not. Our technique uses the inverse of an attention mask to generate a noise estimate that is then used to denoise temporal observations. We apply this to the task of camera-based physiological measurement. A convolutional attention network is used to learn which regions of a video contain the physiological signal and generate a preliminary estimate. A noise estimate is obtained by using the pixel intensities in the inverse regions of the learned attention mask, this in turn is used to refine the estimate of the physiological signal. We perform experiments on two large benchmark datasets and show that this approach produces state-of-the-art results, increasing the signal-to-noise ratio by up to 5.8 dB, reducing heart rate and breathing rate estimation error by as much as 30%, recovering subtle pulse waveform dynamics, and generalizing from RGB to NIR videos without retraining.

## 1 INTRODUCTION

Attention mechanisms have been successfully applied in many areas of machine learning and computer vision (Mnih et al., 2014; Vaswani et al., 2017), including object detection (Oliva et al., 2003), activity recognition (Sharma et al., 2015), language tasks (Anderson et al., 2018; You et al., 2016), machine translation (Bahdanau et al., 2014), and camera-based physiological measurement (Chen & McDuff, 2018). An additional benefit of attention mechanisms is that they are interpretable and show which regions of an image were used to generate a particular output. In this paper, we focus on a counter-intuitive question – is there important information contained within the regions that are typically ignored by the attention models? And, can we exploit information in this region to improve the quality of estimation for the underlying signals of interest?

We focus on the specific temporal prediction problem of camera-based physiological measurement as an exemplar application for our approach. The SARS-CoV-2 (COVID-19) pandemic has rapidly changed the face of healthcare, emphasizing the need for better technology to remotely provide care to patients. COVID-19 is linked to serious heart and respiration related symptoms (Xu et al., 2020; Zheng et al., 2020; Puntmann et al., 2020). Even after the COVID-19 crisis, many doctor appointments could be carried out online with telemedicine technology, increasing the flexibility for appointments. Recent research in computer vision has led to the development of non-contact physiological measurement techniques that leverage cameras and computer vision algorithms (Takano & Ohta, 2007; Verkruysse et al., 2008; Poh et al., 2010a; De Haan & Jeanne, 2013; Wang et al., 2017; Chen & McDuff, 2018). Camera-based vital signs could also enable driver monitoring (Nowara et al., 2018), face anti-spoofing (Liu et al., 2020a; Nowara et al., 2017), or long-term human-computer-interaction (HCI) studies (McDuff et al., 2016) where wearing contact devices for extended periods may be infeasible. Convolutional networks currently provide state-of-the-art performance on heart rate (HR) and breathing rate (BR) measurement from video (Chen & McDuff, 2018; Yu et al., 2019; Liu et al., 2020b).

While the convolutional neural networks may be able to accurately learn what features in the image are important for finding the physiological signals, they may not be able to learn a good model of the noise that corrupts the signals. The noise present in the video, which is considered to be "everything else than the signal of interest", may be caused by many diverse factors and could vary greatly across videos and datasets. Possible sources of noise include changes in head motion (Estepp et al., 2014),

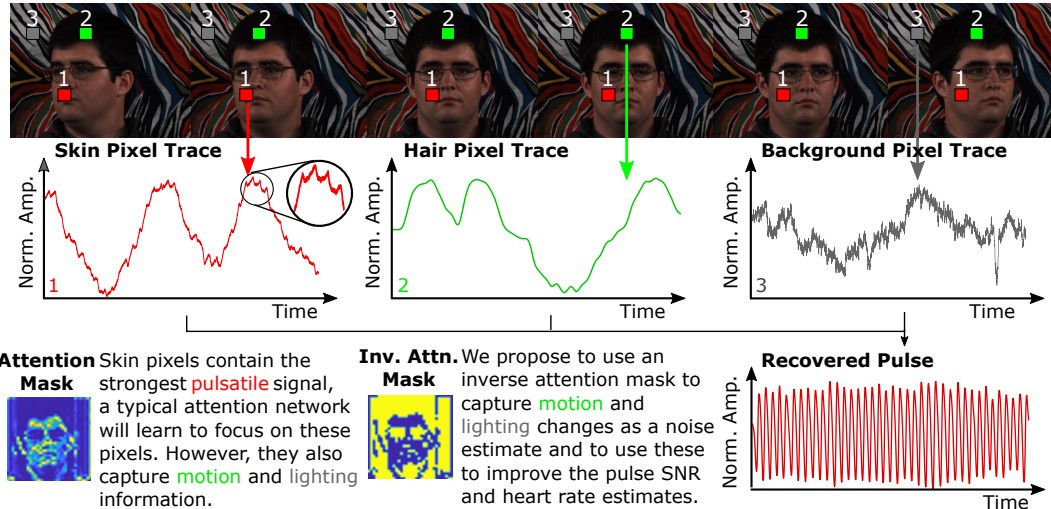

Figure 1: We propose an approach, using the regions ignored by the attention mechanism (e.g., hair or background), as noise estimates. We train a model to learn a denoising mapping to remove these noise sources from the physiological signal. Our approach produces more accurate physiological waveforms, even in severely challenging scenarios.

facial expressions (Zhang et al., 2016), speech, ambient light variations (Nowara et al., 2018), and video compression artifacts (Yu et al., 2019; Nowara & McDuff, 2019). The wide variety of possible noise sources makes it challenging for any model to explicitly capture a good noise representation and to remove that noise from the signals of interest.

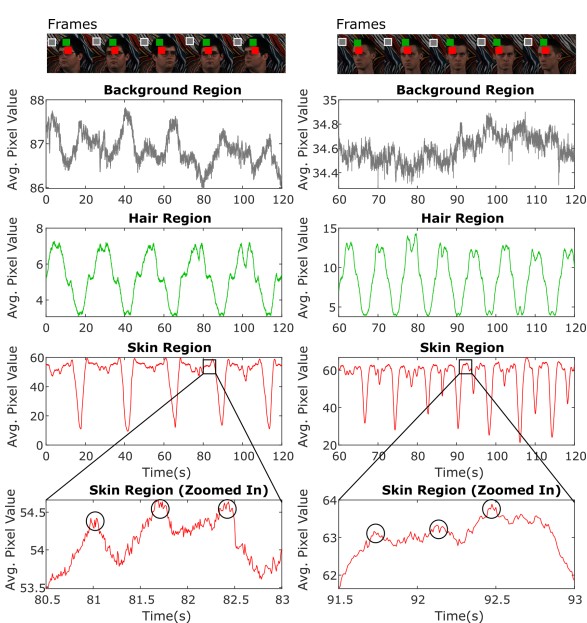

Figure 2: Temporal changes of background (gray), hair (green) and skin (red) pixel intensities in a video are correlated due to large change (e.g., motion, shadows). Physiological signals are very subtle (see zoomed in portion). This motivates us to use the inverse attention region to denoise the physiological estimates.

The key observation we make is that regions ignored by an attention mechanism in a neural model likely contain information about sources of noise that are also present in the regions used by the attention mechanism to compute the physiological signals. Using the "distraction" regions that were ignored by the attention masks offers a way to estimate the noise for each video without making any assumptions about the nature of the noise. The only assumption is that most regions not used by the attention masks do not contain the signals of interest and consequently contain noise.

We demonstrate that we can use the intensity variations from regions outside of the attention mask as a noise estimate and learn a denoising mapping to remove noise from the recovered signals. See Fig. 1 for an overview of our denoising approach. Fig. 2 shows additional examples of pixel traces from different regions in the frame, demonstrating that the regions ignored by the attention masks often contain noise correlated with the noise in the selected skin regions. We show that our approach outperforms state-of-the-art methods on several datasets across a range of HR and BR error measures. Our denoising approach also generalizes well to new data, even data recorded with different imaging modalities,

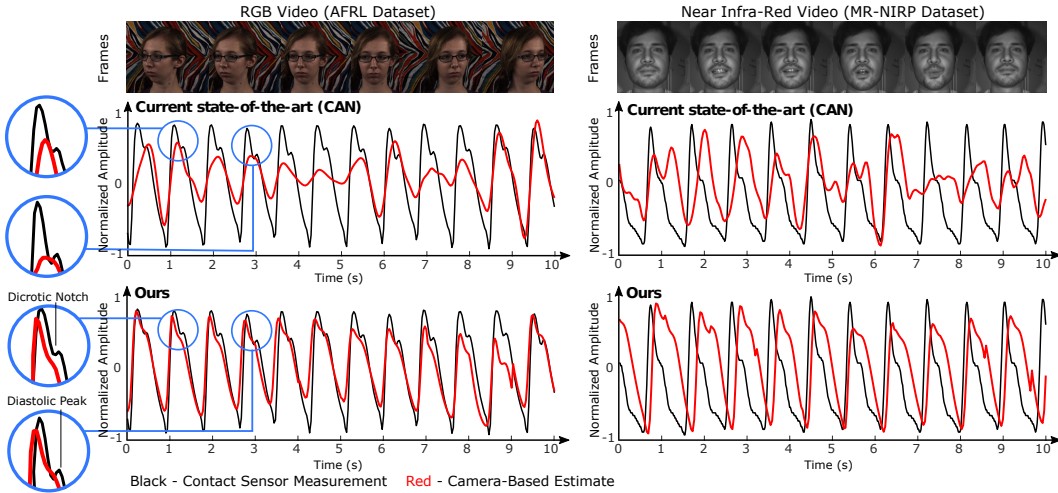

Figure 3: Pulse signals output by a state-of-the-art network and our denoising method. Our method produces cleaner signals, free from motion artifacts (present in the benchmark method), and better matching the ground truth subtle dynamics and shape. Notice the zoomed-in portions with easily identifiable dicrotic notch and diastolic peaks in our outputs.

such as near-infrared (NIR), without any additional training. Our proposed approach is also able to recover very subtle waveform dynamics, such as the clearly visible dicrotic notch, shown in Fig. 3, which is challenging for video-based methods. Obtaining clean and more accurate waveforms is useful for determining important health metrics, such as blood pressure (Elgendi et al., 2019), which is infeasible with current methods. The idea of using the inverse attention regions is likely very useful in a wide variety of vision tasks, ranging from activity recognition to deblurring. However, in this work, we focus on physiological measurement due to the clinical importance.

The core contributions of this paper are to: (1) propose the use of inverse attention masks for generating noise estimates, (2) present a novel method for denoising non-contact physiological measurement using this approach, (3) evaluate our method on three datasets showing state-of-the-art performance on pulse and respiration measurement, (4) demonstrate that our approach generalizes to NIR data without further training. Supplementary material including code, models, video examples and additional experimental results are provided with this submission.[1]

## 2 RELATED WORK

**Attention Mechanisms.** Attention mechanisms provide a way for a model to learn which parts of an image or video "are relevant for the task at hand and attach a higher importance to them" (Sharma et al., 2015). During training the attention weights are learned reflecting the importance of the embedding features. Recently, transformer models, based solely on attention mechanisms, have become popular (Vaswani et al., 2017). In convolutional neural networks (CNNs) these attention mechanisms typically form a spatial mask. These masks can help practitioners understand the decision-making process of a network (Fukui et al., 2019) and in certain cases the "fixations" of attention generated by computer models and by human observers were very similar (Oliva et al., 2003). Attention mechanisms can be used to connect layers; for example, one which focuses on temporal information (e.g., trained on flows) and another which focuses on spatial information (e.g., trained on RGB frames). Prior work has found that these crosslink layers guide the spatial-stream to pay more attention to the human foreground areas and can be less affected by background clutter (Tran & Cheong, 2017). In physiological measurement, two-layer networks have been found to be effective as both color and motion information are valuable for extracting the subtle physiological signal in the presence of noise (Chen & McDuff, 2018). While attention mechanisms often work well, they are a simple representation of which regions are important. However, pixels outside these regions may provide useful context or a strong prior about the noise present.

**Physiological Imaging.** Volumetric changes in blood over time lead to subtle changes in light reflected from the skin and subtle motion variations which can be measured with a camera (Takano

---

[1]https://github.com/AnonymousCodeSubmission/Benefit_of_Distraction

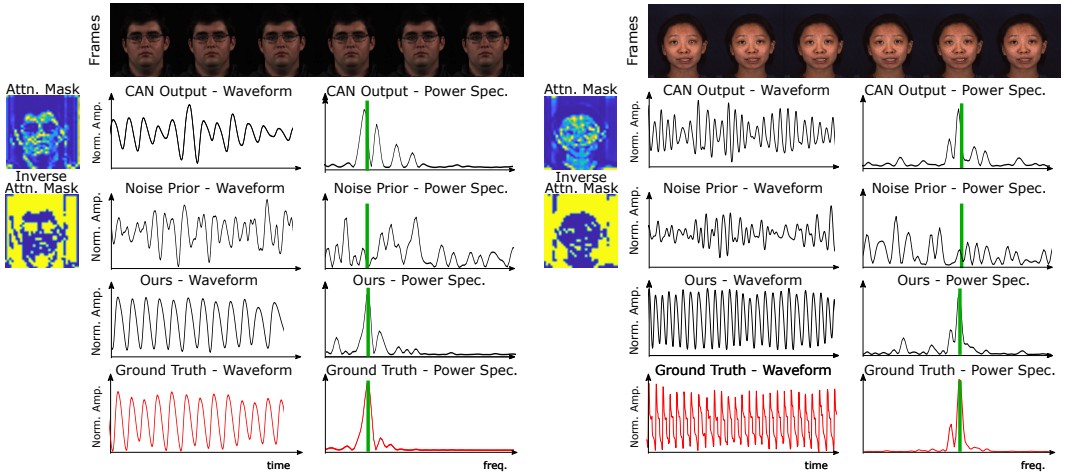

Figure 4: Examples of attention masks, inverse attention masks (yellow=higher weight) and the corresponding original physiological estimates, the noise estimates (shown for the green channel only for visual clarity), and denoised physiological signals from AFRL (left) and MMSE-HR (right) datasets. Green lines indicate the ground truth HR.

& Ohta, 2007; Verkruysse et al., 2008). The physiological signal obtained from a video can be used to recover several metrics and vital signs, including heart rate (Poh et al., 2010a), heart rate variability (Poh et al., 2010b), breathing rate (Poh et al., 2010b), blood oxygenation (Tarassenko et al., 2014) and pulse transit time (Shao et al., 2014). NIR (Nowara et al., 2018; Chen et al., 2018) and thermal cameras have also been successfully used for measuring physiological signals in the dark (Garbey et al., 2007; Pavlidis et al., 2016). Unfortunately, the signals of interest in video-based physiological measurement are often very subtle and can be easily corrupted by noise due to body motions and ambient lighting changes. Early work in physiological imaging used properties of the physiological signal, e.g., the periodic nature (Poh et al., 2010a) and hemoglobin absorption spectra (De Haan & Jeanne, 2013; Wang et al., 2017) to recover the underlying physiological signal via de-mixing methods (Li et al., 2014; Macwan et al., 2019; 2018; Tulyakov et al., 2016). Some of these unsupervised methods make simple assumptions that the pulse signal should be periodic (non-Gaussian) and that any other source signals are noise (e.g., ICA). Others, such as POS, assume that the plane orthogonal to skin contains the pulsatile signal and non-orthogonal planes contain specular reflections and noise. Others have used physical skin models to learn a mapping from color changes (McDuff et al., 2018) in these noise is not modeled explicitly. Recently, several groups have demonstrated that deep learning models free from heuristic assumptions about the signal structure can perform better, especially in presence of large motion and noise (Chen & McDuff, 2018; Zhan et al., 2019; Špetlík et al., 2018; McDuff, 2018; Niu et al., 2018a;b; Yu et al., 2019; Lee et al., 2020). These end-to-end methods do not explicitly define the noise but rather learn to recover the physiological signal in a fully supervised manner. We show that the performance of a state-of-the-art model is significantly improved by using the distraction regions as an explicit noise estimate.

## 3 BENEFITING FROM DISTRACTION

Let us take a video of a person moving as an example. The skin pixels will contain information about the physiological signal but they will also capture the body motion, as in each frame the incident light changes with the orientation of the head (see Fig. 1). In contrast, the hair pixels will not contain information about the physiological signal (as there are no blood vessels in the hair) but will still contain information about the motion. In this section, we explain how we use those inverse attention (or "distraction") regions to denoise the physiological signals. The details of the proposed deep learning architecture are provided in Fig. 5.

### 3.1 PHYSIOLOGY AND NOISE ENCODER

The backbone of the encoder is formed using a convolutional attention network (CAN) (Chen & McDuff, 2018). This contains appearance and motion branches learned jointly through an attention mechanism. The appearance model is trained directly on the input video frames. It learns from the color and texture information which regions in the video are likely to contain strong physiological

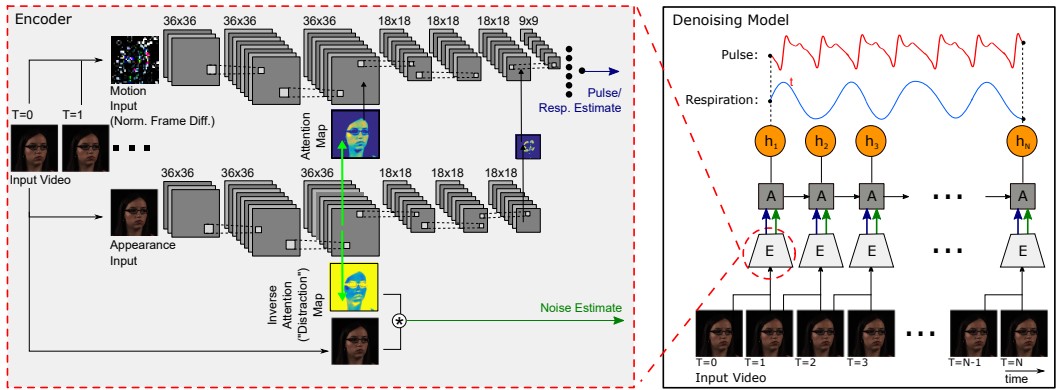

Figure 5: Proposed denoising architecture. The encoder provides the initial physiological signal and the noise estimates to the LSTM at each time step which outputs a denoised physiological signal.

signals. The motion model is trained on the difference of two consecutive video frames to differentiate between the intensity variations present in the video caused by the characteristic physiological variations from those from other sources. The attention mask then reflects a heatmap of the strength of the pulsatile signal in each region of the frame. As shown in the first row of Fig. 4, the attention masks mostly focus on skin regions known to have strong physiological signals, while ignoring other regions, such as the eyes, hair, and background regions. The CAN normally outputs a single one-dimensional (1D) physiological signal estimate. However, we perform an element-wise multiplication of the original input frame with the inverse of the attention mask weights to compute a secondary noise estimate.

We compute the noise signals at each time step by multiplying the inverse attention masks with each channel of each video frame in an element-wise manner. We then spatially average the resulting weighted pixel intensities to obtain the noise estimate:

$$N_{c,t} = \frac{1}{H}\frac{1}{W}\sum_{x=1}^{H}\sum_{y=1}^{W} I_{x,y,t} \circ M_{x,y,t} \tag{1}$$

where $I_t$ and $M_t$ are the frame and mask at time $t$. $N_{c,t}$ is the noise estimate from each [R, G, B] camera channel $c$ at time $t$, and H and W are the image height and width, respectively. The attention and the inverse attention masks were $34 \times 34$ pixels and the video frames were downsampled to the same size using bicubic interpolation.

We normalize the attention mask elements to a range between 0 and 1. To obtain a noise estimate, we set all values larger than a fixed threshold, T, to 0 and everything else to 1, creating a binary mask. Based on the experiments we found a threshold of 0.1 worked well. This binary inverse attention mask ignores regions in the video initially used to compute the physiological signals and keeps all other regions. Examples of inverse attention masks and the corresponding noise estimates are shown in the second row of Fig. 4.

## 3.2 DENOISING MODEL

Our denoising model is then formed as long short-term memory (LSTM) network with the encoder providing input at each time step. The goal is to learn a denoising function to further clean the physiological estimates. As input to the denoising LSTM we stacked the physiological signal and noise signal outputs generated by the encoder. The contact physiological signal (e.g., finger pulse oximeter) was used as ground truth for training. The noise estimates guide the LSTM to learn which waveform features are related to noise and which are related to the physiological signal of interest. The LSTM learns to suppress the noise from the physiological signal and outputs a cleaner waveform matching the ground truth physiological signal better (see the third row of Fig. 4). See the video in the supplementary material for more examples of noise estimates and denoised signals.

We used a two-layer bidirectional LSTM, with 128 hidden units, trained for 10 epochs with Adam optimizer (Kingma & Ba, 2014) and MSE loss. Because the LSTM tends to work better on shorter

sequences, we split each video to sequences of 60 samples, with 50% overlap between time windows, which corresponded to two seconds for the 30 frames per second (fps) videos. Physiological datasets are often relatively small due to the complexity associated with collecting carefully synchronized physiological signals and high-quality videos. Therefore, we implemented the CAN and the denoising LSTM as two separate networks to reduce the number of training parameters.

## 4 DATASETS

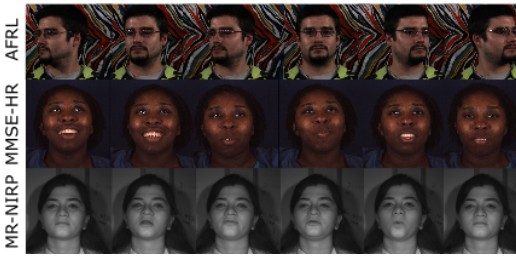

Figure 6: Examples of images used to evaluate our proposed approach.

We evaluated our approach on the following two RGB video datasets and an NIR video dataset.

**AFRL** (Estepp et al., 2014): 300 videos of 25 participants were recorded as $658 \times 492$ pixel images at 120 fps. Fingertip reflectance photoplethysmograms (PPG), electrocardiograms (ECG), and respiratory signals were recorded as ground truth signals. We used the ECG signals to compute the HR estimation errors, the PPG signals to train the network for estimating HR, and respiratory signals for computing the errors and training the network for BR estimation. Each participant was recorded 12 times in each five minute experiment with varying motion and two different backgrounds. The participants: 1) sat still and rested their chin on a headrest, 2) sat still without the headrests, 3) moved their head horizontally at a speed of 10 degrees/second, 4) 20 degrees/second, 5) 30 degrees/second, 6) reoriented their head randomly once every second. We center-cropped the ARFL video frames to $492 \times 492$ pixels to remove the blank background areas.

**MMSE-HR** (Zhang et al., 2016): 102 videos of 40 participants were recorded at 25 fps capturing $1040 \times 1392$ resolution images during spontaneous emotion elicitation experiments. Ground truth blood pressure (BP) wave was measured at 1000 fps and average HR updated after every heart beat. We used the blood pressure waves to train the network and the average HR to compute the HR estimation errors. 19 videos had erroneous average HR estimates, so we recomputed them by using the BP waveform. We detect peaks in the blood pressure waveform and compute the interbeat interval (IBI) between the detected peaks. Heart rate is estimated as $\frac{1}{\mu(IBI)}$ where $\mu(IBI)$ is the mean IBI.

**MR-NIRP (NIR)** (Nowara et al., 2018): Eight participants were recorded with a NIR camera. The videos were recorded at $640 \times 640$ resolution and 30 fps. Fingertip transmission photoplethysmograms were recorded as ground truth signals. Each participant was recorded twice, once sitting still and once performing motion tasks involving talking and randomly moving the head. Because the background in MR-NIRP was not uniform, we applied face detection in the first video frame and cropped a rectangular region with 110% width and height of the detected bounding box.

## 5 TRAINING DETAILS

**Training the Encoder.** Due to the large number of parameters we pretrain the encoder on the largest dataset (AFRL (Estepp et al., 2014)) and lock the weights. When training the encoder the loss is calculated as the mean squared error between the physiological estimate and the ground truth. We do not compute the loss using the noise estimate as there is no ground truth noise signal. In our experiments we performed training and testing separately for each of six motion tasks from the AFRL dataset with a participant-independent cross-validation, leaving out 20% of the participants in each validation split. For experiments on the MMSE-HR and MR-NIRP datasets we used the trained model from Task 2 as these contained the most similar amplitude motions. To maximize the diversity of the participants that this model was trained on to improve the generalizability to new datasets, we instead used a subject-dependent cross-validation, using four minutes of each video for training and one minute for testing.

**Training the Denoising Model.** When evaluating on the AFRL dataset we trained the denoising model with the same subject-independent procedure as for the encoder on AFRL. The MMSE-HR dataset has fewer videos than the AFRL dataset; therefore, we used a leave-one-subject-out cross-validation where we left out all videos of one subject and trained the model on all remaining videos,

repeating this for each subject. The MR-NIRP dataset was small and not suited for training the networks, so we used the LSTM trained on the AFRL dataset. This allowed us to test cross-dataset generalization.

We bandpass filtered ([0.7 Hz, 2.5 Hz]) and detrended the signals (Tarvainen et al., 2002). We normalized the signals by subtracting the temporal mean, dividing by the temporal standard deviation in each video, and normalized their amplitudes to -1 and 1. We resampled all sequences to 30 fps. The signals from each video were divided into 30-second non-overlapping windows. We evaluated the performance of our proposed denoising approach using mean absolute error (MAE), root mean square error (RMSE), Pearson's correlation coefficient ($\rho$) between the estimated HR and the ground truth HR, SNR of the estimated physiological signals (De Haan & Jeanne, 2013), and waveform mean absolute error (WMAE) computed between the estimated and the ground truth signal. See the supplementary material for the definitions of the error metrics.

## 6 RESULTS AND DISCUSSION

We compare four variants of our proposed approach to four state-of-the-art methods for recovering the pulse signal (Chen & McDuff, 2018; Poh et al., 2010a; De Haan & Jeanne, 2013; Wang et al., 2017) and two methods for recovering the breathing signal (Chen & McDuff, 2018; Tarassenko et al., 2014) (see the supplementary material for implementation and signal pre-processing details). The variants of our approach we compare are: training our model with noise estimates as input ("Distraction") and without noise estimates as input ("No Noise"). We can also directly subtract the noise estimate from the signal estimate either in the time domain - "Wave. Sub.", or compute the power spectrum of the estimated noise and signal and subtract the noise spectrum from the signal spectrum - "Freq. Sub.".

**Heart Rate Estimation.** Our method achieves lower HR MAE, RMSE and waveform MAE and higher HR correlation ($\rho$) and SNR (see Tables 1) compared to previous approaches on two large datasets. On the AFRL dataset the MAE is reduced from 2.93 beats per minute (BPM) to 2.25 BPM (25% reduction in error), and on the MMSE-HR dataset the MAE is reduced from 3.74 BPM to 2.50 BPM (33% reduction in error). This shows that information excluded by the attention mask can be successfully leveraged to remove noise, leading to substantial improvements in signal quality. Moreover, the proposed denoising approach is able to recover the subtle waveform dynamics, reducing the waveform MAE by more than 50% on MMSE-HR. While simply subtracting the noise from the signals in the frequency domain often improved the SNR, it did not improve the heart rate estimates. Subtracting the noise signal in the time domain performed even worse and had a particularly negative impact on the BVP SNR. All results were statistically significant (p < 0.01) – see supplementary material for F-test results.

Table 1: Heart rate (HR) and breathing rate (BR) estimation on 3 public datasets. Including the "distraction" regions significantly improves the HR and BR estimation.

| | Heart Rate | | | | | | | | | | | | | | | Breathing Rate | | | | |
| | AFRL | | | | | MMSE-HR | | | | | MR-NIRP(NIR) | | | | | AFRL | | | | |
| Method | MAE | RMSE | SNR | $\rho$ | WMAE | MAE | RMSE | SNR | $\rho$ | WMAE | MAE | RMSE | SNR | $\rho$ | WMAE | MAE | RMSE | SNR | $\rho$ | WMAE |
|---|---|---|---|---|---|---|---|---|---|---|---|---|---|---|---|---|---|---|---|---|
| Distraction | 2.25 | 5.68 | 6.44 | 0.87 | 0.21 | **2.27** | **4.90** | **5.00** | **0.94** | **0.19** | 2.34 | 4.46 | 2.27 | **0.85** | 0.45 | **2.44** | **4.23** | **14.20** | **0.35** | 0.28 |
| No Noise | **2.12** | **5.37** | **6.86** | **0.88** | **0.21** | 2.80 | 6.36 | 4.30 | 0.90 | 0.21 | 2.56 | 5.23 | **2.28** | 0.80 | 0.40 | 2.49 | 4.26 | 14.06 | 0.34 | **0.27** |
| Freq. Sub. | 2.92 | 6.67 | 3.66 | 0.82 | 0.24 | 3.97 | 9.93 | 4.49 | 0.76 | 0.57 | 8.58 | 17.59 | -4.56 | -0.11 | **0.31** | 5.03 | 7.45 | 7.78 | 0.12 | 0.31 |
| Wave. Sub. | 2.92 | 6.66 | 3.09 | 0.82 | 0.24 | 6.09 | 10.84 | -4.75 | 0.71 | 0.55 | 8.83 | 17.0 | -4.69 | -0.17 | **0.31** | 7.21 | 8.83 | 4.12 | -0.03 | 0.37 |
| MAICA | – | – | – | – | – | 3.91 | – | – | 0.86 | – | – | – | – | – | – | – | – | – | – | – |
| RhythmNet | – | – | – | – | – | – | 5.49 | – | 0.84 | – | – | – | – | – | – | – | – | – | – | – |
| PVM | – | – | – | – | – | 4.38 | – | – | 0.82 | – | – | – | – | – | – | – | – | – | – | – |
| CAN | 2.93 | 6.69 | 3.36 | 0.82 | 0.23 | 4.06 | 9.51 | 0.63 | 0.77 | 0.57 | 7.78 | 16.8 | -3.24 | -0.03 | 0.36 | 4.86 | 7.32 | 8.33 | 0.10 | 0.27 |
| POS | 4.36 | 9.45 | 0.73 | 0.74 | 0.45 | 3.90 | 9.61 | 2.33 | 0.78 | 0.39 | – | – | – | – | – | – | – | – | – | – |
| Tulyakov | – | – | – | – | – | – | 11.37 | – | 0.71 | – | – | – | – | – | – | – | – | – | – | – |
| Li | – | – | – | – | – | – | 19.95 | – | 0.38 | – | – | – | – | – | – | – | – | – | – | – |
| Tarassenko | – | – | – | – | – | – | – | – | – | – | – | – | – | – | – | 3.68 | 5.52 | -6.22 | 0.29 | 0.29 |
| CHROM | 4.07 | 9.72 | 0.29 | 0.72 | 0.41 | 3.74 | 8.11 | 1.90 | 0.82 | 0.37 | – | – | – | – | – | – | – | – | – | – |
| ICA | 5.78 | 11.8 | 0.42 | 0.58 | 0.43 | 5.44 | 12.0 | 3.03 | 0.66 | 0.42 | – | – | – | – | – | – | – | – | – | – |

MAICA (2019) = (Macwan et al., 2019), RythmNet (2019) = (Niu et al., 2019), PVM (2018) = (Macwan et al., 2018), CAN (2018) = (Chen & McDuff, 2018), POS (2017) = (Wang et al., 2017), Tulyakov (2016) = (Tulyakov et al., 2016), Li (2014) = (Li et al., 2014), Tarassenko (2014) = (Tarassenko et al., 2014), CHROM (2013) = (De Haan & Jeanne, 2013), ICA (2010) = (Poh et al., 2010a).

**Breathing Rate Estimation.** In addition to estimating heart rate, which is based on intensity variations in the skin, our method can also be used to estimate breathing rate (BR) which is based on motion variations and is more challenging in presence of body motions. Only the AFRL dataset (Estepp et al., 2014) had gold standard reference breathing signals, therefore we were not able to evaluate

our BR results on the other datasets. Our method achieves a reduction in MAE from 3.68 BPM to 2.44 BPM (a 34% error reduction) over the baselines and an increase in SNR of 5.87 dB (Table 1). Our method also obtains cleaner breathing signals compared to the baseline (Fig. 7).

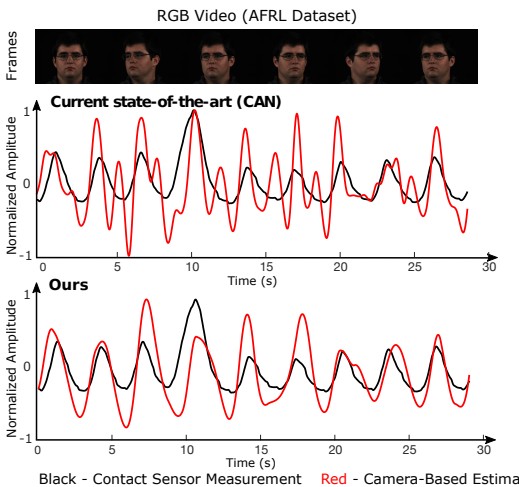

Figure 7: Respiration signals output by a state-of-the-art network and our denoising method. The signals produced by our method are cleaner and match the ground truth better.

**True Benefit of Distraction Regions.** Using our model without noise estimates from the output of the CAN to the LSTM works well when the signals do not change much over time, and when the noise in the training and test sets is similar, e.g., training and testing on AFRL (Table 1). However, including the distraction regions yields improvements in both HR and BR estimates when the signal varies over time or there is a large domain gap between training and testing sets. For example, distraction regions improve the performance on MMSE-HR which has sudden pulse variations, uncontrolled motion, and the presence of facial expressions and on the more challenging NIR MR-NRIP dataset (Table 1). Moreover, including the distraction regions improves the HR and BR estimation accuracy when we train our model only on stationary videos of AFRL (Task 1) and test on videos with large random motions (Task 6) (Table 2). SNR is often higher in the "no noise" condition because it simply produces a smoother signal leading to greater sparsity in the frequency domain. However, the

dominant frequency of the signal (used to compute HR and BR) is often erroneous, resulting in higher MAE and RMSE, and lower $\rho$. These results show that the distraction signal is useful above and beyond including a temporal component to the model.

Table 2: Training on AFRL Task 1 and testing on Task 6. The ignored regions help when the training and test set are very different.

| Method | Heart Rate | | | | | Breathing Rate | | | | |
|---|---|---|---|---|---|---|---|---|---|---|
| | MAE | RMSE | SNR | $\rho$ | WMAE | MAE | RMSE | SNR | $\rho$ | WMAE |
| Distraction | **5.29** | **9.33** | -2.07 | **0.70** | 0.32 | **4.28** | **6.00** | 5.93 | **0.10** | 0.34 |
| No Noise | 5.61 | 9.72 | **-1.91** | 0.67 | 0.32 | 4.38 | 6.15 | **5.96** | 0.07 | 0.34 |

**Transfer learning.** NIR videos of MR-NIRP are more challenging than RGB because the physiological signal is an order of magnitude weaker in the NIR range compared to the visible range, making it very prone to motion artifacts. When trained solely on RGB videos (AFRL dataset) without any fine-tuning, our method outperforms all the baselines across all

five metrics on the NIR videos from the MR-NIRP dataset. As shown in Table 1 the MAE drops from 7.78 BPM to 2.34 BPM (70% reduction in error). Other baseline methods require multiple color channels and therefore cannot be applied to NIR videos.

**Varying Head Motion.** We performed an analysis of the performance on motion tasks of AFRL (Estepp et al., 2014) (see Table 3). Our method shows improvements over the baseline methods on videos across all head motions. For instance, on videos with an angular head rotation of 30 deg/sec (Task 4) the HR MAE was reduced from 2.82 BPM to 1.94 BPM (30% reduction in error) and BR MAE was reduced from 4.85 BPM to 2.88 BPM (40% reduction in error).

**Inverse Mask Definition.** We tested computing the inverse attention mask in two different ways. The first, as a matrix of continuous values in which each element of the inverse mask $M$, $M_{i,j}$, was $1 - A_{i,j}$ where $A$ is the attention mask weights normalized from 0 to 1. The second approach was to threshold these values to create a binary mask where $A'_{i,j} = 1$, if $A_{i,j} > $T. Where T is a threshold from 0 to 1. We found we obtain comparable results with binary (2.25 BPM) or continuous inverse attention masks (2.10 BPM). We also found that the results were not very sensitive to the value of T (see supplementary materials).

**Importance of Different Distraction Regions.** Certain regions in the video may contain more useful information about the sources of noise than others. For example, regions closer to the face may contain more information about the motion of the participant, while regions farther in the background may contain more information about other sources of noise, such as illumination changes. We compared separately using noise estimates from distraction regions closer to the face (center of the frames) and further from the face (edges of the frames). When motion was small, all regions contributed similarly to denoising (MAE = 1.08 BPM with center regions and MAE = 1.07 BPM with edges). But when there was large head motion, regions close to the head (center of the frames) helped the most (MAE = **6.53** BPM with center regions and MAE = 8.74 BPM with edges). See supplementary materials for detailed results.

Table 3: Motion increasing from 1 to 6 on AFRL

| Method | \multicolumn HR 1 | 2 | 3 | 4 | 5 | 6 | BR 1 | 2 | 3 | 4 | 5 | 6 |
|---|---|---|---|---|---|---|---|---|---|---|---|---|
| | **Heart Rate MAE** | | | | | | **Breathing Rate MAE** | | | | | |
| | 1 | 2 | 3 | 4 | 5 | 6 | 1 | 2 | 3 | 4 | 5 | 6 |
| Distraction | **1.06** | 2.11 | **1.79** | **1.94** | 2.50 | 4.78 | **1.42** | **1.86** | 1.88 | **2.88** | **2.87** | **4.15** |
| No Noise | 1.14 | 1.90 | 1.80 | 3.39 | **2.04** | **4.52** | 1.47 | 1.95 | **1.68** | 2.96 | 2.99 | **4.15** |
| Freq. Sub. | 1.52 | 2.62 | 2.51 | 3.00 | 2.58 | 5.30 | 4.30 | 5.35 | 4.89 | 5.27 | 5.09 | 5.26 |
| Wave. Sub. | 1.57 | 2.59 | 2.53 | 3.03 | 2.72 | 5.09 | 4.31 | 5.24 | 4.88 | 5.17 | 5.08 | 5.19 |
| CAN | 1.52 | 2.61 | 2.51 | 3.00 | 2.62 | 5.34 | 4.24 | 5.17 | 4.58 | 5.09 | 4.92 | 5.15 |
| POS | 1.42 | **1.52** | 2.84 | 3.86 | 6.33 | 10.16 | – | – | – | – | – | – |
| CHROM | 1.33 | 1.62 | 2.87 | 2.82 | 3.91 | 11.86 | – | – | – | – | – | – |
| ICA | 2.18 | 2.64 | 4.74 | 4.93 | 7.02 | 13.18 | – | – | – | – | – | – |
| Tar. | – | – | – | – | – | – | 2.51 | 2.53 | 3.19 | 4.85 | 4.22 | 4.78 |

**Effect of Glasses.** Interestingly, our method performed best on subjects who wore glasses. The attention masks for subjects with and without glasses were comparably good. However, CAN performed worse on subjects with glasses and our approach offered a large improvement on those videos (MAE [BPM] with glasses: Ours = **2.17**, CAN = 3.33, and without glasses: Ours = **2.55**, CAN = 2.57). See supplementary materials for example attention masks and results.

## 7 CONCLUSION

We have presented a novel approach for generating noise estimates from inverse attention masks to improve camera-based physiological signal measurements. We hypothesized that the noise affecting regions used by the attention masks to compute the signal of interest would likely be present in other regions in the video which are ignored by the attention masks. Our proposed denoising method outperformed all state-of-the-art methods in heart rate and breathing rate estimation from videos. The recovered BVP signals are also sufficient to recover subtle waveform dynamics present in the ground truth contact signals, including the dicrotic notch and the diastolic peak. Our approach trained on RGB videos showed strong cross-dataset and cross-modality generalizability, outperforming the existing methods on challenging NIR videos.

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
