# OpenReview forum: "The Benefit of Distraction: Denoising Remote Vitals Measurements Using Inverse Attention"
_ICLR.cc/2021/Conference — Reject_

### Official Review · AnonReviewer3 · 2020-10-27
**This paper studied the video based remote physiological signal measurement problem, and proposed a method to modeling the noises w.r.t. the physiological signal using inverse attention mask. The method is evaluated on three dataset, and compared to a few baselines.**

**Rating:** 4
**Confidence:** 5

**Review:**

1. It is interesting to model the noises mixed with the very weak physiological signal; however, the assumption used in this paper, i.e., most regions outside of the attention masks do not contain the signals of interest and consequently contain noise does not really hold, because, besides the weak physiological signal, the dominant signal inside the face video is the face content, i.e., the texture of the face describing the appearance of a person; these background signal dominant the video but cannot be treated as noises.

2. The novelty of the proposed method is quite limited. The general idea is to use an inverse mask, i.e., {1- attention mask}, to determine the regions less correlated with physiological signal, within which the average intensity is computed and treated as noise. However, as we know that for Gaussian-like noises, averaging the local pixel is to remove the noise instead of obtaining the noise. Therefore, the claimed method to compute noise is not well motivated and explained.

3. The experiments are performed on three datasets; however, the authors seem to not follow the widely used protocols but defined a new protocol instead. As a result, even though there a number of SOTA methods in this topic as listed in the references, they were not used to compare with the proposed approach. So it is not clear how the proposed approach can advance the methodology. In addition, while several large physiological signal datasets are available, e.g., with 100-500 subjects, the authors did not use these datasets for evaluations.

4. It is not clear how the authors can measure an output wavelet match the ground-truth BVP signal? particularly when there is misalignment between the video and the finger pulse oximeter wave.

---

> ### Author Response · Authors · 2020-11-12
> **Thank you for your helpful review. We are glad you found our approach interesting and we are grateful for the detailed feedback about the novelty, details of our experiments and our evaluation protocol.**
>
> Thank you for your helpful review. We are glad you found our approach interesting and we are grateful for the detailed feedback about the novelty, details of our experiments and our evaluation protocol.
>
> 1. The noise we are addressing in this work is temporal and can be caused by various factors, such as motion, illumination variations or video compression. We are not addressing spatial noise, such as sensor noise or the texture or intensity of the face, which are usually not the dominant sources of noise in this domain. There are many sources of temporal noise which are likely present in multiple regions in the video, including the face and non-face pixels. For example, the hair region likely contains the same motion variations which are present on the skin, and background regions will likely contain similar ambient light variations or compression artifacts. We have created a new figure (Fig. 2) in the manuscript with additional examples of pixel traces from different regions in the frame to illustrate that the regions ignored by the attention masks often contain noise correlated with the noise in the selected skin regions. Moreover, the attention masks do not focus entirely on all pixels on the face, only the regions with strong signals. It often happens that there are ignored regions inside the face region which contain skin but do not seem to contain a strong physiological signal. In those cases, our method can also improve the performance by using ignored regions as noise estimates. Our approach significantly outperforms the state-of-the-art methods on three benchmark datasets and generalizes to NIR data without additional training. To our knowledge this is the first work to use regions ignored by attention masks for denoising temporal predictions.
>
> 2. It is true that Gaussian noise can easily be reduced by averaging and there are many denoising methods when the noise model is known. However, the noise addressed by our approach is usually not Gaussian, and has sudden and large temporal variations. Therefore, spatial averaging does not remove this noise. We spatially average multiple pixels to obtain the noise and signal estimates to reduce the subtle camera sensor noise but this is not the dominant noise in our problem. The idea of using the ignored regions for the noise estimate is motivated by treating noise as ``any variation not related to the signal of interest". Following this definition, it is intuitive to use the regions which do not contain the signal of interest as representations of noise. While it is true that not all ignored regions in the image will contain the noise that is present in the regions of interest, very likely some of the ignored regions will contain it. However, even when we use all ignored regions to estimate the noise, we achieve significant improvements in the results, we argue that this is strong empirical evidence that our approach is well motivated.
>
> 3a. We used five error metrics commonly used in the literature (see Wang et al. 2017, Niu et al. 2018b, etc.) to evaluate our results and we compared our results to five existing state-of-the-art methods. We used a standard training and testing procedure (see Chen et al. 2018) and we wanted to test cross-dataset generalization, so we also present results on different datasets. We have included results on the MMSE-HR dataset with additional published methods for comparison in Table 1. Could you clarify if there is something else about our protocol that you are referring to?
>
> 3b. With the exception of VIPL-HR which has 107 subjects, we are using the largest available datasets. AFRL has 300 videos of 25 subjects and MMSE-HR has 102 videos of 40 subjects. The other existing physiological datasets that we are aware of (PhysioNet) do not contain videos and therefore do not apply.
>
> 4. To compute the waveform mean absolute error (WMAE) we align the ground truth and estimated signals by using cross-correlation. However, the datasets we used already had well synchronized ground truth signals and the misalignment is very small.

---

### Official Review · AnonReviewer4 · 2020-10-28
**review 351**

**Rating:** 5
**Confidence:** 4

**Review:**

This paper presents a new method for the camera based physiological measurement task. The key idea of this work is to use attention mechanism to learn discriminative features from regions of interest, and use the inverse attention mask to select contextual information to learn noise representation for refinement. Experiments on three datasets show state-of-the-art performance.

I have several main concerns.

1. The main idea of using the reverse attention mask to learn noise information is not significant. In fact, using reverse attention to focus on other regions has been studied in other areas (e.g., for saliency detection). The paper applies the idea to a specific task, i.e., the camera based physiological measurement task.

2. The motivation is not well justified. It is not explained the advantages of using reverse attention mask to learn noise representation. Taking Figure 1 for example, it seems that the attention mask focuses on some key facial regions. While these regions of interest also contain noisy informaiton (e.g., motion and lighting change), the reverse attended regions (the textured background) may not necessarily contain. Why not directly learn an attention map for noise estimatation, or why not directly use the original attention map to learn noise representation?

3. The discussion of how existing methods deal with large motion and noise is not given in the related work section. It is then hard to evaluate the significance of proposed method.

4. The computation of noise (Eq.1) is a multiplication of the reverse attention mask and the original input. However, I do not understand how this operation can be interpreted as noise estimation, as it seems to be a region selection process. Given that, the denoising model seems to be a refinemet model that considers reverse attended regions and the output of the first model. It is hard for me to understand the second model as a denoising model.

Minor issues.

1. The description of Fig.1 can be improved with more details.

2. It is better to provide more details of how errors are corrected in the MMSE-HR dataset (Last sentence of the first paragraph in page 6).

3. Section 5 talks about the training details. It would be better to directly use "Training Details" instead of "Experiments".

4. In the first paragraph of section 6, four variants are given. However, it is not easy to understand the last two, i.e., how the noise subtraction is performed in frequency or time domain.

5. Tables 3 -> Table 3 (page 8).

6. The abstract mentions two datasets but there are actually three datasets in the experiments.

---

> ### Author Response · Authors · 2020-11-12
> **We thank you for the detailed feedback, especially the comments regarding the novelty and motivation for our approach.**
>
> We thank you for the detailed feedback, especially the comments regarding the novelty and motivation for our approach.
> 1. To our knowledge this is the first work to use regions ignored by attention masks for denoising temporal predictions. Our results demonstrate that including the "distraction" regions is effective and improves the results above and beyond temporal modeling alone. We show that our approach outperforms the state-of-the-art methods on three benchmark datasets and generalizes to NIR data without additional training. We focus on physiological measurement, but this approach could generalize to other applications using attention networks. We believe that the deep learning community will benefit from the insights gained from our experiments.
> 2. The reason why we do not explicitly learn the noise representations from the videos is that the noise sources can be very diverse and it is often impossible to faithfully model such variable noise. Therefore, a model trained to estimate noise on one set of videos may not generalize well to datasets with different noise sources. Moreover, we do not have ground truth noise measurements that could be used for supervision. The idea of using the ignored regions as a noise estimate is motivated by treating noise as "any variation not related to the signal of interest". Following this definition, it is intuitive to use the regions which do not contain the signal of interest as representations of noise. While it is true that not all ignored regions in the video will contain the noise that is present in the regions of interest, some of these ignored regions will likely contain it. For example, the hair region usually contains the same intensity variations caused by motion which are also present on the skin (see Fig. 1). Similarly, background regions will likely contain similar ambient light variations or compression artifacts. We have created a new figure (Fig. 2) in the manuscript with additional examples of pixel traces from different regions in the frame to illustrate that the regions ignored by the attention masks often contain noise correlated with the noise in the selected skin regions. We found that different ignored regions contain different sources of noise and contribute differently to denoising (Section 6). However, even when we use all ignored regions to estimate the noise, we achieve improvements in the results, we argue this is empirical evidence that our approach is well motivated.
> 3. This is a very helpful suggestion. We have added additional material to the related work to clarify how existing methods deal with large motion and other noise. Some methods use demixing approaches with simple assumptions that the pulse signal is periodic (non-Gaussian) and that any other source signals are noise (e.g., ICA). POS assumes that the plane orthogonal to skin contains the pulsatile signal and non-orthogonal planes contain specular reflections and other noise. While others use an end-to-end supervised deep learning framework where the noise is not defined explicitly, such as the cited CAN method.
> 4. It is correct that our noise estimation step is equivalent to region selection. We find regions in the video which do not contain the signal of interest (the ignored regions). Some of those ignored regions likely contain noise similar to the noise in the regions with the signal of interest. To obtain temporal noise estimates from the ignored regions we spatially average the pixel intensities and pre-process the signals (Section 3.1). Then, we treat the noise estimates as an additional feature input to the supervised learning framework along with the original noisy estimates of the physiological signal. The network can use the noise estimates to denoise the physiological signals and match the ground truth signal better.
>
> Minor issues:
> 1. Thank you, we updated the figure caption with more details.
> 2. We detect peaks in the blood pressure waveform and compute the interbeat interval (IBI) between the detected peaks. Heart rate is estimated as 1/mean(IBI). We have added this description to the main text.
> 3. Thank you, we renamed the section to "Training Details".
> 4. We obtain noise estimates as time signals. We can either directly subtract the noise estimate from the signal estimate in the time domain, or compute the power spectrum of the estimated noise and signal and subtract the noise spectrum from the signal spectrum. However, we find that simply subtracting the noise from the signal does not remove all artifacts. Instead, we train an LSTM network to learn a denoising mapping to remove the noise estimates from the estimated signals to obtain cleaner physiological signals.
> 5. Thank you, we have corrected "Tables 3" with "Table 3".
> 6. We used two large public datasets and a small dataset with NIR videos. We mention the first two large datasets in the abstract and only mention that our method also performs well on the NIR dataset with transfer learning.

---

### Official Review · AnonReviewer1 · 2020-10-28
**Interesting insight in the drawbacks of attention**

**Rating:** 9
**Confidence:** 4

**Review:**

The article proposes a model for estimating physiological signals from videos. The novelty of the proposed approach is to use the low attention regions to estimate a model of the signal noise. Using this estimate can improve the performance of the denoising component.

The model is well described and sensible and the experimental section demonstrates state-of-the-art results on what is a very relevant task.

Overall, the article describes an interesting approach and contains valuable insights into the limitations of attention-based approaches in noisy signals.

---

> ### Author Response · Authors · 2020-11-12
> **We are glad you found our work interesting and valuable. We appreciate the comments that our model is well described and that the experimental approach is sensible.**
>
> We are glad you found our work interesting and valuable. We appreciate the comments that our model is well described and that the experimental approach is sensible.

---

### Decision · Program_Chairs · 2021-01-07
**Final Decision**

**Decision:**

Reject

**Comment:**

The paper introduces a procedure that uses low attention areas to de-noise temporal prediction. The paper appears to focus on 'temporal noise' as opposed to constant noise present in the video (it may handle shifting shadows, but not background noise).

The idea is certainly interesting, however, the experimental protocol suffers from the issues pointed out by reviewer 3:
- maintaining the same protocol as prior methods to ensure a direct comparison of the results against reported scores by the sota
- in the context of attention, alignment (or lack or it) is extremely important; assuming perfect alignment is not very realistic
(if the alignment is perfect, one might try a simple method such as taking all readings at a point over time and considering the mode, then correcting any outliers in the off-attention areas)

These specific issues were not fully addressed during the review period.

The questions raised by reviewer 2 were addressed to a satisfactory degree in the rebuttal.